# Physically Ground Commonsense Knowledge for Articulated Object Manipulation with Analytic Concepts

## Abstract

We human rely on a wide range of commonsense knowledge to interact with an extensive number and categories of objects in the physical world. Likewise, such commonsense knowledge is also crucial for robots to successfully develop generalized object manipulation skills. While recent advancements in Multi-modal Large Language Models (MLLMs) have showcased their impressive capabilities in acquiring commonsense knowledge and conducting commonsense reasoning, effectively grounding this semantic-level knowledge produced by MLLMs to the physical world to thoroughly guide robots in generalized articulated object manipulation remains a challenge that has not been sufficiently addressed. To this end, we introduce analytic concepts, procedurally defined upon mathematical symbolism that can be directly computed and simulated by machines. By leveraging the analytic concepts as a bridge between the semantic-level knowledge inferred by LLMs and the physical world where real robots operate, we are able to figure out the knowledge of object structure and functionality with physics-informed representations, and then use the physically grounded knowledge to instruct robot control policies for generalized, interpretable and accurate articulated object manipulation. Extensive experiments in both simulation and real-world environments demonstrate the superiority of our approach. **Please refer to the appendix for more details, and our codes will be made publicly available.**

## 1 Introduction

Studies in human cognition (Carey & Xu, 2001; Leslie et al., 1998; Ullman, 2000; Biederman, 1987; Hummel & Biederman, 1992) figure out that humans possess a wide range of commonsense knowledge which guides us to properly interact with an extensive number and categories of objects in the physical world. Such knowledge spans multiple physical concepts, including the spatial structures, functional properties and physical dynamics of objects. We are able to ground the commonsense knowledge from human cognition to the physical world and effectively manipulate real-world objects based on this knowledge to accomplish specific tasks.

Therefore, this commonsense knowledge is also essential for the successful development of general robot manipulation. As recent advancements in Multi-modal Large Language Models (MLLMs) have shown substantial potential in utilizing commonsense knowledge and performing commonsense reasoning (Achiam et al., 2023; Hurst et al., 2024; Liu et al., 2024a; Touvron et al., 2023), researchers have leveraged MLLMs in articulated object manipulation tasks to pursue more generalized robot manipulation capabilities (Li et al., 2024; Tang et al., 2023; Huang et al., 2024; Hong et al., 2023). LLM-instructed paradigm incorporates an understanding of the semantics underlying the manipulation tasks. This facilitates an intuitive comprehension of the task goal, and enabling the semantic-level reasoning between actions and outcomes, which guide the robot to reasonably complete manipulation tasks, especially for open-ended tasks in real-world environment (Bender & Koller, 2020; Huang et al., 2023).

However, since LLMs function at the semantic level, significant challenges still remain in effectively integrating their commonsense reasoning capability into the control policy, which operates at the physical level to manage real-robot interactions. On one hand, encoding knowledge in natural

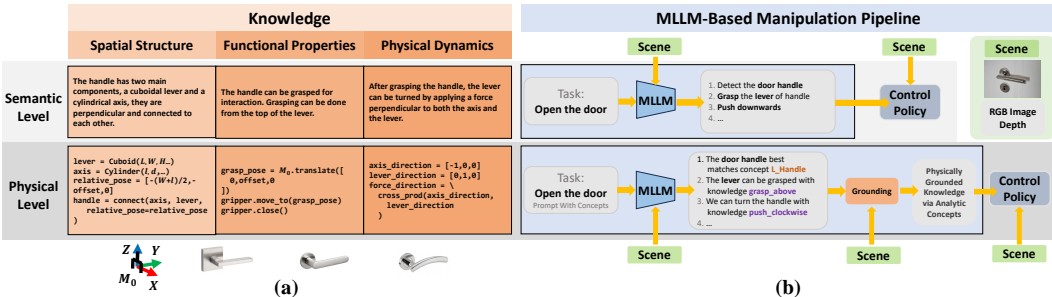

Figure 1: **(a):** An illustration of knowledge represented using both semantic and physical expressions. Compared to semantic expressions, physical ones provide precise mathematical definitions and numerical values, which robots can more effectively interpret and compute for object manipulation (detailed in Fig. 2-Left). **(b):** Illustrations of the conventional MLLM-driven object manipulation pipeline (Upper) and our approach (Lower). Using analytic concepts, we bridge the gap between semantic-level representations, which MLLM excels at, and the physical world, on which robot operate. Please refer to Fig. 2 for descriptions of **L_Handle**, **grasp_above** and **push_clockwise**.

language as feature inputs for the control policy may hinder the control policy from fully recognizing the physical concepts underlying the knowledge (Majumdar et al., 2023; Cohen et al., 2024). On the other hand, LLMs struggle with high-precision numerical analysis (Hendrycks et al., 2021; Hu et al., 2024), making it difficult to fine-tune an LLM to convey commonsense knowledge in a sufficiently precise physical form for manipulation tasks that demand high accuracy. Fig. 1-(a) illustrates the differences between knowledge in the semantic form learned by LLMs and the physical form understood by robots.

In order to construct a bridge between the semantic-level knowledge inferred by LLMs and the physical world where real robots operate, we introduce analytic concepts. In our vision, a piece of commonsense knowledge encapsulates the essential commonality shared by a group of similar entities. From this perspective, an analytic concept is procedurally defined with mathematical symbols to represent such generalized commonality in a physical form which can be directly computed and simulated by machines. By aligning the semantic knowledge provided by LLMs with analytic concepts and mapping the analytic concepts to the physical world, we can identify important priors regarding object structure and functionality according to these concepts, and finally use these pieces of physically grounded knowledge to instruct robot control policies.

Fig. 1-(b) illustrates our improvement on current LLM-instructed paradigm using analytic concepts. Given an open-ended description of an object manipulation task in natural language and an RGB-D workspace image, a robot should perform appropriate physical interactions. In common practice (Fig. 1-(b)-Upper), a multi-modal LLM (MLLM) analyzes the task description and the image, reasons about where and how to interact with the target object, and outputs a semantic-level task plan to guide the control policy. In contrast (Fig. 1-(b)-Lower), we provide the MLLM not only with the task description and image but also with our proposed analytic concepts. With their designs (decribed in Sec. 3), the MLLM can thoroughly understand and integrate these analytic concepts with other information through its commonsense reasoning capability. We then prompt the MLLM to provide responses by employing analytic concepts, focusing on three critical aspects: the target part for manipulation, its structural knowledge, and the manipulation knowledge required to complete the manipulation task. This approach aligns the semantic-level knowledge inferred by the MLLM with the analytic concepts, which are then grounded in the physical world. The control policy subsequently incorporates the knowledge within the grounded concepts with the RGB-D data of the target object to produce reliable interaction strategies guided by the embedded physical knowledge.

By bridging the semantic knowledge and the physical world through analytic concepts, our approach greatly benefits from the capability of MLLMs in commonsense reasoning and instructing robot control policies for generalized, interpretable, and accurate articulated object manipulation. We have conducted exhaustive experiments in both simulation and real-world environments to demonstrate the superiority of our approach in articulated object manipulation.

In summary, our contributions are as follows: **1)** We introduce analytic concepts to build a bridge between semantic-level knowledge inferred by MLLMs and the physical world where robots operate. **2)** We propose a pipeline to ground semantic-level knowledge in the physical world through analytic

concepts, enabling concrete and precise guidance for robots in completing various manipulation tasks. **3)** Benefiting from MLLMs' strong reasoning capability and analytic concepts as a physically grounded representation, our method demonstrates great superiority in articulated object manipulation tasks across extensive object categories in both simulation and real-world environments.

## 2 RELATED WORKS

### 2.1 ARTICULATED OBJECT MANIPULATION

Articulated object manipulation requires embodied agents to interact with articulated objects using both visual perception and physical reasoning (Mo et al., 2021; Geng et al., 2023; Li et al., 2024; Huang et al., 2024; Qian et al., 2024; Huang et al., 2025). Researchers have approached this problem from multiple aspects. Xiang et al. (2020) builds a physics-rich simulator with diverse real-world object model dataset. Where2Act (Mo et al., 2021) introduces a framework to predict action likelihoods and generate manipulation proposals on a per-pixel basis, facilitating targeted interactions. Where2Explore (Ning et al., 2024) introduces a few-shot learning framework that measures affordance similarity across categories, enabling migration of affordance knowledge to novel objects. GAPartNet (Geng et al., 2023) proposes the idea of generalized actionable parts and leverages the GAPart pose as a kind of structured representation to define heuristic interaction policies for object manipulation. ManipLLM (Li et al., 2024) takes a step further to use the robust reasoning capabilities of LLMs to infer action policies directly from an RGB image featuring a target object. A3VLM (Huang et al., 2025) leverages MLLM to infer the actionable part's bounding box, axis, and semantics, and integrates the information to infer the action policies. These studies have contributed significantly to the development of the field.

### 2.2 MULTI-MODAL LARGE LANGUAGE MODELS

Recent advanced LLMs, such as LLaMa (Touvron et al., 2023) and GPT-3 (Floridi & Chiriatti, 2020), exhibit remarkable abilities in solving complex tasks, highlighting their potential for learning and applying commonsense knowledge. To better harness LLMs' capabilities in vision research, Multi-modal Large Language Models (Achiam et al., 2023; Hurst et al., 2024; Liu et al., 2024a; Lin et al., 2023) have been developed, greatly expanding visual understanding and processing. For example, GPT-4o (Hurst et al., 2024) leverages commonsense knowledge and performs commonsense reasoning to interpret and respond to multi-modal tasks with a nuanced understanding, achieving competitive results in many vision tasks such as image captioning and visual question answering.

However, applying MLLMs to object manipulation tasks (Li et al., 2024; Qian et al., 2024; Huang et al., 2024), which involves precise interactions with the physical world, remains in its early stages. One of the most challenging problem is to align the representations in semantic and visual modalities with the action modality of a robot which concretely grounds in the physical world (Ma et al., 2024). In this paper, we propose analytic concepts to better leverage the power of MLLMs in the physical world and achieve more generalized robotic manipulation skills.

## 3 ANALYTIC CONCEPTS

In this section, we introduce how analytic concepts are developed to represent commonsense knowledge in a physical form. As shown in Fig. 2-Left, each analytic concept comprises three components, concept identity, analytic structural knowledge, and analytic manipulation knowledge. Since manipulation knowledge is linked with a specific spatial structure, we focus primarily on structural knowledge when designing a concept and integrate the related manipulation knowledge within it. **Detailed examples of analytic concept implementation are provided in the appendix.**

### 3.1 CONCEPT IDENTITY

First of all, we should give an identity to a concept as a unique symbol for a piece of commonsense knowledge, ensuring one-to-one correspondence. Besides, we further introduce a precise and concise synopsis of this concept along with the identity, so that the concept can be clearly understood not only by humans but also MLLMs.

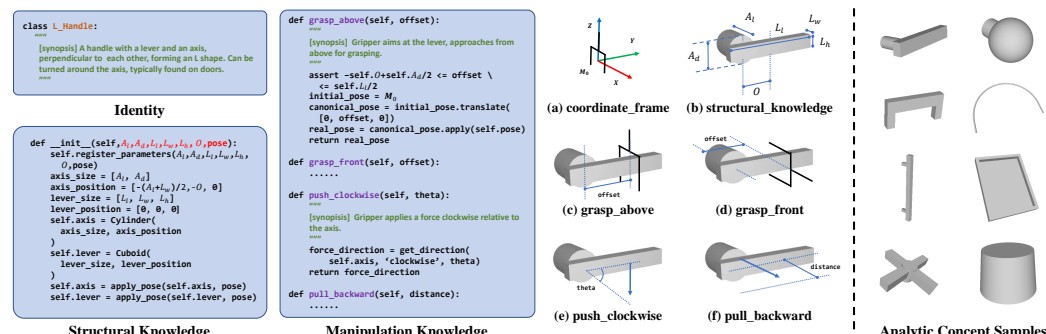

Figure 2: [Left] Example implementation of concept `L_Handle`, including **Concept Identity**, **Analytic Structural Knowledge** defined in 3D space (a-b), and **Analytic Manipulation Knowledge** (c-f). [Right] Visualizations of representative analytic concepts.

## 3.2 ANALYTIC STRUCTURAL KNOWLEDGE

The analytic structural knowledge is a series of mathematical procedures revealing the essential commonality of the spatial structure, including spatial layout and structural relationships, shared by all instances of the concept. Further, there are variable parameters in the procedures to represent the variations among different physical instances. That is, a physical instance of this concept can be created with specific parameters, and in turn, a target in the physical world can be also resolved into parameters of a concept.

## 3.3 ANALYTIC MANIPULATION KNOWLEDGE

To properly interact with a spatial structure, we should know about its functional properties like affordances and physical dynamics like the effect of a force on it. From this point of view, we provide manipulation knowledge of grasp poses that meet the functional properties of the concept and force directions that would cause effective movement. Similar to the analytic structural knowledge, analytic manipulation knowledge is also formulated by mathematical procedures with variable parameters. A concept may contain multiple pieces of manipulation knowledge, as shown in Fig. 2-Left-(c-f), where each piece is represented by a function that can be parameterized into a specific grasp pose or force direction. Such design of manipulation knowledge covers all atomic actions applicable to the analytic concepts, and achieves diversity through the variation of parameters. We also provide a precise and concise synopsis for each piece of manipulation knowledge to help MLLMs perform better reasoning on them.

## 3.4 DISCUSSIONS

**Implementation.** As each analytic concept is a symbol with procedural definitions, it is convenient and scalable to implement them as class templates, where inheritance enhances scalability in construction and promotes code reusability. Specifically, Python scripts is used for implementation.

**Creation of Analytic Concepts.** As analytic concepts represent commonsense knowledge widely used in our daily life, such knowledge is easily understood, simply and clearly defined, and highly generalizable. We invite volunteers with high-school level math skills to create analytic concepts, as well as knowledge and corresponding synopses of their created analytic concepts, which are checked by experts for correctness. We figure out that a concept along with knowledge and synopses can be finished in less than two hours on average and the same concept created by different volunteers show a high degree of consistency. This indicates that the analytic concepts can be easily created, possess great robustness and show strong consensus among different people. So far, we have created 153 analytic concepts and thanks to their strong generalization capability, we find that only a small portion of them are sufficient to handle a large number of daily object manipulation tasks. Fig. 2-Right provides the visualizations of representative analytic concepts. We will make our created analytic concepts publicly available to facilitate the research of the community.

## 4 METHODOLOGY

In this section, we discuss about our methodology of grounding semantic-level knowledge reasoned by MLLMs in the physical world with analytic concepts, and how physically grounded representations can benefit articulated object manipulation. **More details and illustrations are provided in the appendix.**

**Problem Formulation.** We study articulated object manipulation under following setting. Given an RGB image and its depth map featuring an object, as well as a task description in natural language, a robot is required to conduct proper physical interactions with the object with a parallel gripper to complete the task. This requires an intelligent system capable of commonsense reasoning to understand the object and task description, and also the capability of managing the complex physics involved in generating affordances and strategies for contact-rich interactions.

**Pipeline.** Fig. 3 presents the pipeline of our methodology, containing three major steps: i) target the part for manipulation, ii) ground the structural knowledge, and iii) ground the manipulation knowledge. Then, the robot performs interaction with the object guided by the grounded knowledge. We clearly demonstrates how these steps operate in the following sections.

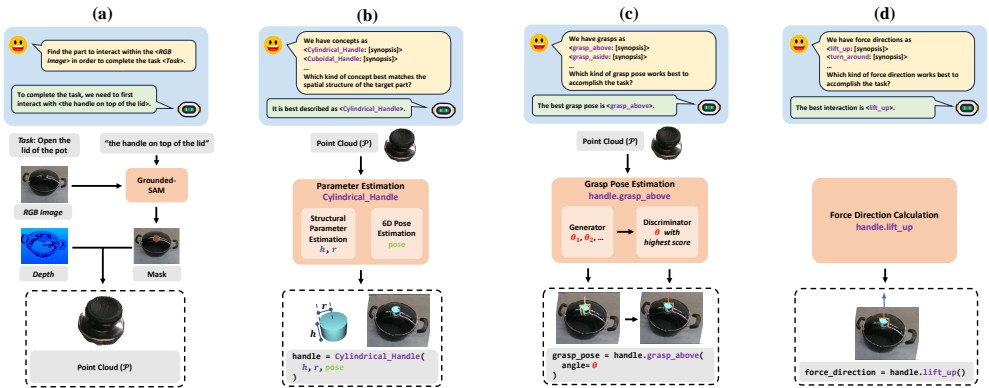

Figure 3: Illustration of our proposed approach with three major steps. (a) Target the part for manipulation. (b) Ground the structural knowledge. (c-d) Ground the manipulation knowledge.

### 4.1 TARGET THE PART FOR MANIPULATION

In the first step, we use an MLLM to identify the target part for manipulation from the task definition and RGB image. Specifically, we adopt GPT-4o (Hurst et al., 2024) as the MLLM and prompt it with "*find the **part** to interact within the <RGB Image> in order to complete the task <Task>, and determine the <category> of the **part***". This yields both a semantic-level description and the category of the target part. The part's description is then passed to Grounded-SAM (Ren et al., 2024), together with the RGB image, to produce a pixel-level segmentation of the part. The segmentation mask is applied to the depth image to crop the point cloud $\mathcal{P}$ of the target part, while the category of the target part is preserved for use in identifying the analytic concept in the next step.

### 4.2 GROUND THE STRUCTURAL KNOWLEDGE

After obtaining the point cloud $\mathcal{P}$ of the target part being manipulated, we further leverage the MLLM to reason about the structural knowledge of this part and ground it on corresponding analytic concept with a parameter estimator.

**Concept Identification.** Leveraging the commonsense reasoning capability of the MLLM, we can identify an analytic concept that best represents the spatial structure of the target part. Specifically, analytic concepts are first categorized into different groups (*e.g.* handle, lid) according to the spatial structure they describe and their synopses. To identify the best-matching analytic concept for the target part, we provide the MLLM with concepts from the same category as the target part, using those concepts' identities along with the synopses in natural language. The MLLM is then prompted to determine "*which **concept** best matches the spatial structure of the target part*". In this manner, we are able to align the semantic-level knowledge embedded in the MLLM with analytic concepts.

**Parameter Estimation.** With the identified concept, we then estimate the parameters of this concept to ground it on the target part. There are generally two types of parameters: i) ones in the formulations of the structural knowledge that controls the spatial structure of analytic concepts, and ii) 6-dof pose parameters that record the global translation and rotation in world coordinates. Therefore, the parameter estimation process is also divided into two steps accordingly.

For structural parameters, we use a point cloud encoder (Zhao et al., 2021), with an average pooling operation to encode the point cloud of the target $\mathcal{P}$ into deep features and then use different MLPs to regress the parameters of each type of analytic concept.

For 6-dof pose parameters, we take $\mathcal{P}$ as input and also use a Point-Transformer encoder and a subsequent MLP decoder as network, while in this case the decoder outputs a point cloud $\mathcal{P}^*$ representing $\mathcal{P}$ in canonical space. Then we apply Umeyama algorithm (Umeyama, 1991) and RANSAC (Fischler & Bolles, 1981) for outlier removal in order to acquire the translation and rotation from $\mathcal{P}^*$ to $\mathcal{P}$. The translation and rotation can be adopted as the 6-dof pose of the analytic concept.

### 4.3 GROUND THE MANIPULATION KNOWLEDGE

So far, we have identified the analytic concept corresponding to the target part and grounded the concept on its real-world point cloud by estimating the parameters. To guide the robot to complete the manipulation task, we still need manipulation knowledge involving grasp pose about how the gripper should grasp the target part and force direction about how the gripper should move after grasping. Similar as grounding the structural knowledge, the first step is to prompt the MLLM with manipulation knowledge of grasp pose / force direction along with corresponding synopsis and ask the MLLM to determine "*which kind of **grasp pose / force direction** works best to accomplish the task*". Then, we discuss how the two types of manipulation knowledge are physically grounded.

**Grasp Pose.** Each piece of analytic manipulation knowledge for grasp pose refers to a category of grasp poses belonging to the same pattern, and an exact grasp pose $\mathbf{G}$ is physically grounded by estimating the parameters of such analytic knowledge. Therefore, we develop a neural network for each kind of grasp pose knowledge to estimate its proper parameters given the target point cloud $\mathcal{P}$.

Different from the structural parameters which are unique for a specific part, there are many possible answers for the grasp pose knowledge parameters, as illustrated in the bottom-left image in Fig. 3-c. To this end, we adopt the idea of conditional GAN (Mirza, 2014) to develop a generative framework. The generator $G$ uses MLPs to generate multiple possible candidates of parameters from a Gaussian noise $z$ conditioned on deep features of $\mathcal{P}$ embedded with a Point-Transformer. Then, we incorporate a discriminator $D$ to score each candidate and select the best one. The discriminator has a Point-Transformer encoder for $\mathcal{P}$ and an MLP encoder for the parameters, concatenates the encoded features together and decodes them with MLPs as a score in $[0, 1]$. With the parameters, a physically grounded grasp pose $\mathbf{G}$ can be calculated according to the analytic manipulation knowledge.

**Force Direction.** After obtaining the grasp pose $\mathbf{G}$, the function for force direction knowledge can be invoked to calculate a vector $\mathcal{F}$ indicating the direction in which the force should be applied to the target part when manipulating the object.

### 4.4 INTERACT WITH THE OBJECT

Guided by the physically grounded knowledge embedded in analytic concepts, a robot can interact with the object heuristically. The robot first moves its gripper to the grasp pose $\mathbf{G}$ and completes the grasping. After that, the gripper moves along the force direction $\mathcal{F}$ to complete the task.

### 4.5 TRAINING AND IMPLEMENTATION DETAILS

**Training Data Preparation.** To prepare data for training the models in our approach, we first provide analytic concept annotations for real objects. Particularly, we annotate the concept parameters for training objects from PartNet-Mobility (Xiang et al., 2020). Then, we place each object in the SAPIEN (Xiang et al., 2020) simulator and use a camera to capture the RGB image and depth map of this object. According to the URDF and analytic concept annotation of an object, we can easily obtain the groud-truth bounding box, point cloud, 6-dof pose and structural parameters of each actionable part. To collect positive and negative samples for grasp pose parameters, we per-

form extensive random sampling for the parameters within its domain, then calculate the grasp pose based on the sampled parameters and test it in the simulator. If the grasp succeeds, the parameter is considered a positive sample; otherwise, it is a negative sample.

**Grounded-SAM.** Grounded-SAM (Ren et al., 2024) consists of two major components, Grounding-Dino (Liu et al., 2024b) and SAM (Kirillov et al., 2023). We keep SAM (Kirillov et al., 2023) frozen and fine-tune Grounding-Dino (Liu et al., 2024b) with RGB images of size $336 \times 336$ with ground-truth bounding boxes of the actionable parts, along with natural language prompt that describes the actionable parts provided by GPT-4o (Hurst et al., 2024).

**Structural Parameter Estimation.** The encoder is a Point-Transformer (Zhao et al., 2021) that extracts 128 groups of points with size 32 from the input with 2048 points and has 12 6-headed attention layers. The subsequent MLP with three layers outputs the structural parameters. The network is trained with L2 loss between the estimated and ground-truth structural parameters.

**6-dof Pose Estimation.** The network shares the same architecture as that used in structural parameter estimation. The network is trained with chamfer distance loss between the estimated point cloud and the point cloud of the input in canonical space which is obtained by transforming the input point cloud according to its ground-truth 6-dof pose.

**Grasp Pose Estimation.** The Point-Transformer and MLPs in both generator and discriminator shares the same architecture as that used in structural parameter estimation. As we already have positive and negative grasp pose parameter samples, we first train the discriminator with $\mathcal{L}_D = -\mathbb{E}_{x \sim p_{\text{data+}}} [\log D(x|y)] - \mathbb{E}_{x \sim p_{\text{data-}}} [\log (1 - D(x|y))]$, where $D(x|y)$ is the output score of the discriminator representing the likelihood of sample $x$ in the data distribution given condition $y$, and $p_{\text{data+}}$ and $p_{\text{data-}}$ denotes the positive and negative data distribution. Then we train the generator with $\mathcal{L}_G = -\mathbb{E}_{z \sim p_z} [\log D(G(z|y))]$ where $G(z|y)$ refers to the generated sample according to noise $z$ given condition $y$, and $p_z$ is a Gaussian distribution.

## 5 EXPERIMENTS

We thoroughly evaluate our approach on articulated object manipulation tasks against five representative approaches (Mo et al., 2021; Geng et al., 2023; Ning et al., 2024; Li et al., 2024; Huang et al., 2025), including state-of-the-art A3VLM (Huang et al., 2025) to showcase the effectiveness of our approach. In the following sections, we present the simulation setting, experiment results and comprehensive analysis. We also carry out the object manipulation experiments with physical robots in real-world environments to provide a more comprehensive and stronger evaluation. **Please refer to the appendix for detailed settings, analysis and ablation studies.**

### 5.1 SIMULATION SETTING

We design our evaluation in order to reflect the capability of an algorithm to accurately guide a robot to interact with articulated objects with a single gripper. Hence, we generally follow the experiment setting of Where2Act (Mo et al., 2021) which is widely used in the research community.

**Data Setting.** Since our manipulation knowledge is developed for single-gripper manipulation, We have collected a total of 972 objects across 15 categories from PartNet-mobility (Xiang et al., 2020) that are suitable for single-gripper manipulation following (Mo et al., 2021). To evaluate the generalization of our approach to novel categories, we divide the categories into a training group with 10 categories and a testing group with 5 categories.

**Experiment Setting.** We adopt the SAPIEN (Xiang et al., 2020) simulator as the simulation environment for our evaluation. In each manipulation simulation, the target object is initially placed at the center of the scene within the simulator. The object's joint pose is randomized, with a 50% chance of being at the closed state and a 50% chance at open state with random motion. An RGB-D camera with known intrinsic parameters stares at the center of the scene and is positioned at the upper hemisphere with a random azimuth $[0°, 360°)$ and a random elevation $[30°, 60°]$. We use a gripper as the end-effector in our main experiments following (Mo et al., 2021; Ning et al., 2024).

**Evaluation Setting.** We use the success rate as our evaluation metric to assess the capability of articulated object manipulation among different approaches. A manipulation is considered successful

if the target joint's movement exceeds 0.01 unit-length or 0.5 relative to its maximum motion range. The success rate is defined as the ratio of successful manipulations to the total number of test trials. We adopt an interaction budget of 5 for each action proposal.

## 5.2 Simulation Evaluation

We compare our approach with four baselines representing three types of frameworks for articulated object manipulation. Vanilla frameworks, such as Where2Act (Mo et al., 2021) and Where2Explore (Ning et al., 2024), learn directly from 2D images or point clouds and output pixel-level affordance maps. The framework leveraging object structures, like GAPartNet (Geng et al., 2023), also takes point clouds as input but proposes the idea of GAPart, treating the 6-dof poses as a kind of structured representation, and adopts the interaction policies predefined on each GAPart. The third type of framework, including ManipLLM (Li et al., 2024) and A3VLM (Huang et al., 2025), leverages the robust reasoning capabilities of MLLMs to infer affordance knowledge from RGB-D image and natural language inputs featuring a target object. In comparison with these previous frameworks, our approach integrates both the MLLM's reasoning capability and analytic concepts as a physically grounded structured representation for object manipulation tasks.

**Main Results.** The performance of different baselines in Tab. 1 highlights the critical role of commonsense reasoning and physically grounded representations in object manipulation tasks. Compared to vanilla frameworks Where2Act and Where2Explore, both ManipLLM and A3VLM enhance one of these two aspects and achieve certain improvements. After fully integrating these two aspects through our approach by grounding commonsense knowledge inferred by MLLMs on analytic concepts, the success rate of manipulation has been boosted, about 15.2% for train categories and 27.1% for test categories compared with A3VLM, demonstrating the superiority of our proposed analytic concepts and framework. The close results on train and test categories also suggest that our approach can effectively handle unseen objects. We attribute this property to the effect of the following two factors. First, our defined analytic concepts can cover ubiquitous common structure and generalize effectively to describe unseen objects. Second, utilizing its strong commonsense reasoning capability, an MLLM can still find the best-match concepts to the target parts of unseen objects according to the concepts' synopses.

Table 1: Evaluation results on manipulation tasks, averaged on categories. All values represent the average success rate in percentage.

| Method | Where2Act | Where2Explore | GAPartNet | ManipLLM | A3VLM | Ours |
|---|---|---|---|---|---|---|
| Train Cats | 26.1 | 26.9 | 29.7 | 32.0 | 36.9 | 42.5 |
| Test Cats | 14.4 | 20.5 | 28.7 | 30.6 | 32.1 | 40.8 |

**LLM with Physically Grounded Analytic Concept.** ManipLLM and A3VLM use LLaMa-Adapter (Zhang et al., 2023), an MLLM, to directly infer the pose of an end-effector of a robot to interact with the target object. Note that a suction end-effector is originally employed by ManipLLM and A3VLM while the parallel gripper is used in our experiment setting, we show the performances of our approach, ManipLLM and A3VLM with both suction and parallel gripper as end-effector on our data setting in Tab. 2. After changing the end-effector from suction to parallel gripper, ManipLLM and A3VLM both suffer a more severe drop in success rate compared with our approach. This highlights the challenge of relying solely on MLLMs to directly perform accurate numerical reasoning to produce precise affordances and interaction policies for manipulation tasks involving gripper-based grasping and interactions. In comparison, our approach is able to ground the semantic-level knowledge inferred by MLLMs through analytic concepts. In this manner, a robot can precisely calculate the affordances and interaction strategies with these physically grounded concepts, and achieves higher success rates, especially for more challenging settings, *i.e.* gripper as end-effector for grasping.

Table 2: Comparison between ManipLLM, A3VLM and our approach on manipulation tasks using gripper and suction as the end effector. All values represent the average success rate in percentage, where the values for training and testing categories are shown as a pair separated by a slash.

| Method | ManipLLM | A3VLM | Ours |
|---|---|---|---|
| Gripper | 32.0 / 30.6 | 36.9 / 32.1 | 42.5 / 40.8 |
| Suction | 58.0 / 57.8 | 65.0 / 60.9 | 68.5 / 63.2 |

## 5.3 REAL-WORLD EVALUATION

**Settings.** We conduct experiments that involve interacting with 8 real-world household objects from different categories. The manipulation tasks involve *open the box*, *switch the handle of the bucket*, *open the door*, *lift the lid of the kettle*, *open/close the laptop*, *lift the lid of pot*, *open the closet of the storage* and *move the lid of the trashcan*. We employ a robot arm with a parallel gripper as its end-effector and utilize a RealSense D415 camera to capture the RGB-D image. We adopt success rate as metric, where we define a successful interaction as one in which the gripper securely grasps the target object and moves correctly to complete the task. Each task is tested for 10 times.

**Quantitative Analysis.** Tab. 3 shows the performance of our approach in real-world experiments, demonstrating high success rates. The comparison with A3VLM further highlights the importance of a physically grounded knowledge representation in accurate and reliable robot manipulation tasks.

Table 3: Results of real-world experiments. All values represent success rate.

| Method | 📦 | 🪣 | 🚪 | 🫖 | 💻 | 🍲 | 🗄 | 🗑 |
|---|---|---|---|---|---|---|---|---|
| A3VLM | 0.70 | 0.40 | 0.50 | 0.60 | 0.70 | 0.60 | 0.70 | 0.60 |
| Ours | 0.90 | 0.60 | 0.70 | 0.80 | 0.90 | 0.80 | 0.80 | 0.70 |

**Qualitative Analysis.** Fig. 4 visualizes the outputs of each step in four real-world manipulation tasks, demonstrating the our approach's ability to accurately locate the target part, physically ground both structural and manipulation knowledge and successfully complete the manipulation tasks. We attribute this success to the effective integration of both the strong reasoning capability of MLLMs and the physically grounded knowledge represented by analytic concepts.

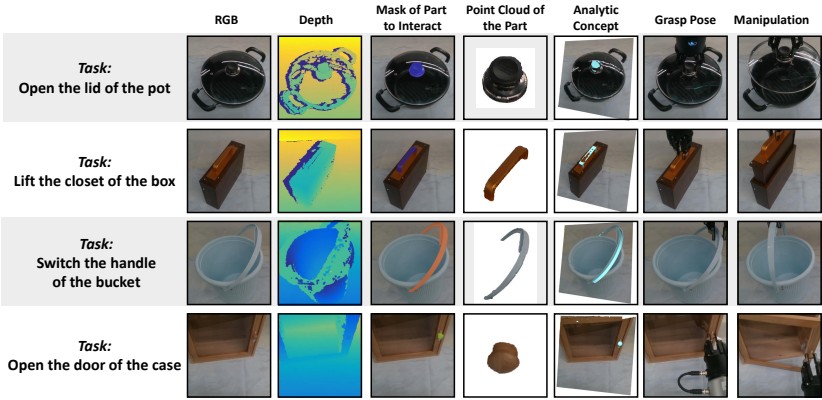

Figure 4: Visualizations of the output in each step in real-world experiments. The leftmost column lists the task description. In the **Analytic Concept** column, light blue meshes represent analytic concepts grounded on the target parts. The **Grasp Pose** and **Manipulation** columns illustrate how the gripper grasps the target parts and completes the manipulation tasks.

## 6 CONCLUSION

In this paper, we focus on articulated object manipulation and introduce analytic concepts to represent commonsense knowledge involved in this task in a physical form. Each concept encompasses an identity, as well as analytic structural and manipulation knowledge procedurally defined with mathematical symbols. These concepts construct a bridge between the semantic-level knowledge and the physical world, helping to overcome the limitations of MLLMs in precise numerical reasoning and the shortcomings of natural language in describing physical concepts. Taking advantages of analytic concepts, we further propose an object manipulation pipeline to successfully translate semantic-level commonsense knowledge inferred by MLLMs into concrete physical-level knowledge, instructing robots to complete various manipulation tasks accurately. Our approach demonstrates its superiority across a wide range of object categories in both simulation and real-world scenarios.

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

# A APPENDIX

This is the appendix of the submission. In the appendix, we provide comprehensive details for better understanding of our main paper and offer more evidence to prove the effectiveness of our approach. The appendix is organized as follows:

A.1: Technical details and discussions about the **implementation of networks** in our approach.

A.2: Details of our main experiments, including **experiment settings**, **experiment results**, and the **analysis of failure cases**.

A.3-A.6: **Ablation studies** and analysis about the **effectiveness of modules and design** in our approach.

A.7: Discussions about the **distillation of analytic concepts** for manipulation tasks.

A.8: Discussions about the **scalability of analytic concepts** on expanding to novel object categories and manual construction.

A.9: Discussions about **detecting and handling error** in manipulation experiments.

A.10: Demonstration of **real-world experiments**, the discussions about the **sim2real gap**, and **robustness** of our approach in real world.

A.11: Details and discussions about the **code implementation of analytic concepts**.

A.12: Discussions about the **generalization capability** of our approach.

A.13: Discussions about **current limitations and the future work**.

A.14-A.15: **Computer resources** and discussions about **societal impacts**.

A.16: Statement on Usage of LLMs.

## A.1 NETWORK STRUCTURE

In this section, we show the structure of the neural networks used in **Structural Parameter Estimation**, **6-dof Pose Estimation** and **Grasp Pose Estimation**.

*Structural Parameter Estimation*

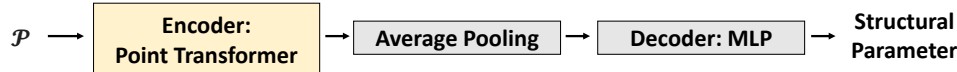

*6-dof Pose Estimation*

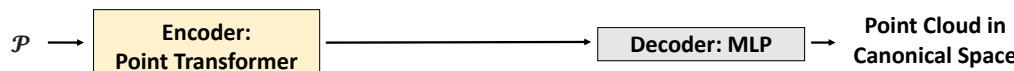

*Grasp Pose Estimation*

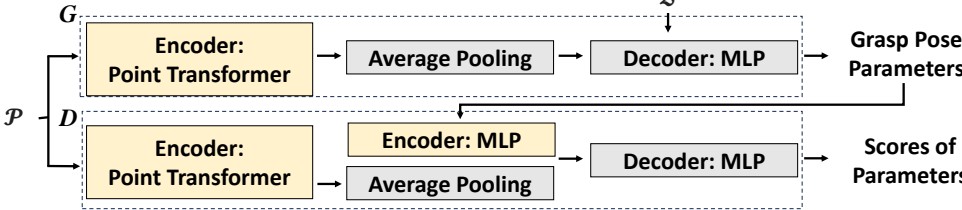

Figure 5: Network structure regarding Structural Parameters Estimation, 6-dof Pose Estimation and Grasp Pose Estimation. $\mathcal{P}$ denotes the input point cloud, and $G, D$ denote the generator and discriminator respectively.

As shown in Fig. 5, the structural parameter estimation network takes the point cloud of the detected part with 2,048 points as input. The it utilizes a Point-Transformer (Zhao et al., 2021) as encoder, which extracts 128 groups of points with size 32 from the input point cloud and has 12 6-headed attention layers. Subsequently, an average pooling layer is introduced to extract the global feature of the entire point cloud. Then, an MLP with three linear layers and accompanied ReLU activation outputs the structural parameters.

For the 6-dof pose estimation network, the Point-Transformer encoder of is directly linked to an MLP that has three linear layers with ReLU activation, which outputs a point cloud of the detected part in canonical space.

The grasp pose estimation network builds on the concept of conditional GAN (Mirza, 2014). The generator $G$ employs Point-Transformer as the point cloud encoder, uses an average pooling layer to aggregate features and generates plausible grasp pose parameters from Gaussian noise $z$ using an MLP decoder. The discriminator $D$ obtains the point cloud feature using a Point-Transformer encoder along with an average pooling layer and encodes the generated grasp pose parameters with MLPs. The two features are then concatenated and passed through another set of MLPs to compute scores for the generated grasp pose parameters.

As our main purpose is to provide a baseline to demonstrate the feasibility and effectiveness of using analytic concepts to physically ground semantic-level commonsense knowledge, we do not delve into complex network designs.

## A.2 EXPERIMENTS

### A.2.1 MANIPULATION EXPERIMENT SETTINGS

**Data Statistics.** In Sec. 5.1 of the main paper, we introduced our data settings for manipulation experiments in simulation environment. Here we provide more details as shown in Tab. 4. We have collected 15 categories of objects from PartNet-Mobility (Xiang et al., 2020), after removing objects that are either too small (*e.g.* Pen, USB), requiring multi-gripper collaboration (*e.g.* Plier, Scissors), or not making sense for robot gripper to manipulate (*e.g.* Fan, Clock). The 15 categories are further divided into two groups, 10 categories for training and the rest 5 categories for testing. The 10 categories involved in training consist of 773 objects, which are further divided into 586 training objects and 187 testing objects. The testing categories consist of 199 objects.

Table 4: Detailed statistics of the data split for manipulation tasks. "–" for training objects indicates that the category is unseen during training.

| Num of Obj | Training Categories | | | | | | | | | | Testing Categories | | | | |
|---|---|---|---|---|---|---|---|---|---|---|---|---|---|---|---|
| | Box | Door | Faucet | Kettle | Microwave | Fridge | Storage | Switch | TrashCan | Window | Bucket | KitchenPot | Safe | Table | Washing |
| Train Obj | 20 | 23 | 65 | 22 | 9 | 32 | 270 | 53 | 52 | 40 | - | - | - | - | - |
| Test Obj | 8 | 12 | 19 | 7 | 3 | 11 | 75 | 17 | 17 | 18 | 36 | 23 | 29 | 95 | 16 |

**Evaluation Settings.** As discussed in Sec. 5.1 of the main paper, we adopt the commonly used metric, success rate, for evaluating manipulation tasks. Following Where2Act (Mo et al., 2021), a manipulation is considered successful if the the target joint on an articulated object moves more than 0.01 unit length or 0.5 relative to its maximum motion range. We use a 5-interaction budget. The success rate is computed as the ratio of successful manipulations to the total number of manipulation trials. 22 analytic concepts are used in our experiments.

We compare our method with five baselines. **1) Where2Act (Mo et al., 2021)** predicts per-pixel action likelihoods and selects the point with highest score. We follow Where2Act to generate 100 action proposals at that point and select the one with highest score for manipualation. The action proposals are generally categorized into pushing and pulling. A task is considered successful if either type of action succeeds. The training data are collected by placing articulated objects in SAPIEN (Xiang et al., 2020) and interacting with them using a two-finger gripper. **2) Where2Explore (Ning et al., 2024)** also predicts actions and proposals on point level, while employs a few-shot framework to enhance its generalization to novel object categories. We use the same training data, test data, and evaluation metric as Where2Act. **3) GAPartNet (Geng et al., 2023)** predicts the 6-dof pose of the target part and defines interaction policies based on the predicted pose. We collect training data according to the GAParts definition in GAPartNet. **4) ManipLLM (Li et al.,**

**2024)** employs an MLLM to predict the contact point on a given RGBD featuring the target object, guided by a language prompt describing the manipulation task. For fair comparison with other approaches, we use a two-finger gripper as the end-effector instead of the originally used suction. **5) A3VLM (Huang et al., 2025)** is the current state-of-the-art method in MLLM-driven affordance learning. It employs an MLLM to predict the 6-dof pose and motion axis of the target part and interacts with it based on the predicted information.

For fair comparison, we use a two-finger gripper as the end-effector instead of the original suction, and further adapt ManipLLM (Li et al., 2024) and A3VLM (Huang et al., 2025) for gripper-based manipulation. For ManipLLM (Li et al., 2024), we collect gripper-compatible affordance maps and conduct hyperparameter search during LLaMa-Adapter fine-tuning and TTA to get the optimal performance. For A3VLM (Huang et al., 2025), it is designed to be end-effector agnostic, therefore no additional tuning of the model is needed. Instead, we propose an independent generic grasp-pose proposer according to A3VLM (Huang et al., 2025) to produce gripper-compatible grasp poses.

### A.2.2 DETAILED EXPERIMENT RESULTS

We provide the detailed results on manipulation tasks for each object category in Tab. 5. The results show that our approach outperforms all baselines across all categories. Notably, for categories consisting of objects with complex structures and multiple joints, such as Table, the success rate of our approach has been boosted 21.4% compared with A3VLM. These results further demonstrate the strength of our approach, which leverages the commonsense knowledge of MLLM to infer the interaction target and manipulation knowledge for complex objects, and grounds the knowledge to physical world using analytic concepts for accurate manipulation. We repeat the experiments for 10 times, and report the error bars of the experiments in Tab. 6.

Table 5: Evaluation results on manipulation tasks across different object categories with four previous baselines. All values represent the average success rate as a percentage. The average success rates are calculated based on the objects. The abbreviations of the object categories are listed as follows: Box, Door, Faucet, Fridge, Microwave, StorageFurniture, Switch, TrashCan, Window, Bucket, KitchenPot, Safe, Table, WashingMachine.

| Methods | Training Categories | | | | | | | | | | | Testing Categories | | | | | |
|---|---|---|---|---|---|---|---|---|---|---|---|---|---|---|---|---|---|
| | Box | Dor | Fct | Fdr | Ket | Mcw | Stf | Swt | Tcn | Win | **AVG** | Bkt | Pot | Saf | Tab | Wsm | **AVG** |
| Where2Act | 6.8 | 31.5 | 17.1 | 31.2 | 9.9 | 31.3 | 37.1 | 19.3 | 17.1 | 12.2 | 26.1 | 8.0 | 6.2 | 20.4 | 17.8 | 9.1 | 14.4 |
| Where2Explore | 9.0 | 38.3 | 16.8 | 35.4 | 11.5 | 33.8 | 34.6 | 23.6 | 19.5 | 15.8 | 26.9 | 14.0 | 10.8 | 31.8 | 22.7 | 15.4 | 20.5 |
| GAPartNet | 15.5 | 45.4 | 17.1 | 40.5 | 11.1 | 35.8 | 40.5 | 18.2 | 19.5 | 13.8 | 29.7 | 21.3 | 14.7 | 30.2 | 37.3 | 12.7 | 28.7 |
| ManipLLM | 11.0 | 50.2 | 16.5 | 36.4 | 10.6 | 41.5 | 47.0 | 20.4 | 15.0 | 15.2 | 32.0 | 19.9 | 12.4 | 37.0 | 39.1 | 19.1 | 30.6 |
| A3VLM | 12.2 | 56.2 | 24.1 | 40.8 | 20.1 | 49.7 | 53.3 | 25.4 | 18.2 | 15.8 | 37.4 | 25.2 | 11.0 | 40.0 | 42.5 | 22.3 | 32.1 |
| Ours | **15.9** | **58.4** | **28.0** | **46.7** | **24.3** | **52.2** | **58.6** | **32.7** | **22.9** | **22.3** | **42.5** | **28.9** | **15.5** | **50.6** | **51.6** | **22.7** | **40.8** |

Table 6: Error bars for the manipulation experiments.

| Methods | Where2Act | Where2Explore | GAPartNet | ManipLLM | A3VLM | Ours |
|---|---|---|---|---|---|---|
| Train Cats | 26.1±0.8 | 26.9±1.2 | 29.7±0.8 | 32.0±1.2 | 36.9±0.9 | 42.5±0.9 |
| Test Cats | 14.4±1.3 | 20.5±0.9 | 28.7±1.0 | 30.6±0.8 | 32.1±0.6 | 40.8±0.7 |

### A.2.3 ANALYSIS OF FAILURE CASES

In most failure cases, we observe that the collision volume of the end-effector is a critical factor. For example, a cylindrical handle attached to a door may allow valid grasps from various directions around its axis. However, in some orientations, the gripper may be obstructed by nearby parts (*e.g.*, the door) when approaching the handle, leading to failed manipulation. In future work, we will take different types and brands of end-effectors into consideration to design more effective action policies for handling such tasks.

### A.3 ANALYSIS OF SYSTEM LIMITATIONS

In this section, we provide a system error breakdown analysis by sequentially replacing the output of each module with ground truth to identify common failure cases and system limitations. Because

ground truth is required for replacement, the analysis is only conducted on training categories. The detailed results are reported in Tab. 7. The experiments demonstrate that the primary bottlenecks in the system are structural parameter estimation and 6-dof pose estimation, where a 20.8% and 14.3% gap on average is observed by replacing these two modules respectively. Errors in estimating an object's spatial structure and 6-dof pose can lead to grasp poses with collisions or misaligned grasps, and preventing the gripper from accurately detecting the target part to complete the manipulation task.

Table 7: Evaluation results of manipulation experiments on system limitations. All values represent the average success rate as a percentage.

| Module | Categories | | | | | | | | | | |
|---|---|---|---|---|---|---|---|---|---|---|---|
| | Box | Dor | Fct | Fdr | Ket | Mcw | Stf | Swt | Tcn | Win | **AVG** |
| None | 15.9 | 58.4 | 28.0 | 46.7 | 24.3 | 52.2 | 58.6 | 32.7 | 22.9 | 22.3 | 42.5 |
| Actionable Part Segmentation | 19.8 | 62.5 | 33.8 | 53.0 | 26.1 | 54.9 | 67.2 | 40.9 | 32.9 | 28.9 | 49.8 |
| Concept Identification | 26.1 | 64.4 | 35.0 | 55.4 | 30.8 | 62.9 | 67.8 | 42.2 | 35.1 | 29.8 | 51.2 |
| Structural Param Estimation | 61.6 | 81.9 | 69.6 | 74.2 | 64.4 | 75.3 | 78.2 | 65.3 | 66.1 | 64.5 | 72.0 |
| 6-dof Pose Estimation | 92.2 | 87.4 | 81.2 | 85.8 | 83.4 | 91.1 | 89.3 | 85.4 | 81.6 | 84.2 | 86.3 |
| Grasp Pose Estimation | 95.6 | 90.2 | 92.5 | 90.6 | 92.2 | 98.5 | 95.3 | 93.5 | 91.8 | 93.0 | 93.6 |
| Force Direction | 99.0 | 99.5 | 98.2 | 99.0 | 98.8 | 99.0 | 98.4 | 96.2 | 99.2 | 99.4 | 98.6 |

## A.4 ANALYSIS OF CONCEPT IDENTIFICATION AND PARAMETER ESTIMATION

As discussed in A.3, structural parameter estimation remains the major bottleneck in overall performance. In addition, concept identification also affects parameter estimation, since parameters align with the identified concept. To further analyze these modules, we provide a detailed quantitative evaluation of both concept identification and parameter estimation, with results shown in Tab. 8.

In the concept identification stage, we provide the MLLM with multiple analytic concept identities along with their corresponding synopses in natural language, prompting it to select the concept that best matches the target part for manipulation. The accuracy of the MLLM's concept identification is measured by comparing its selections to those made by humans. Accurate identification occurs when the concept chosen by the MLLM matches the human-chosen one. The results demonstrate that our concept identity as well as the corresponding synopsis are well-defined to be accurately understood by MLLMs.

The evaluation metric for parameter estimation is the Point2Face distance (Ravi et al., 2020), measured in millimeters. This metric computes the average distance from the points in a point cloud to their closest counterparts on a corresponding mesh. In this context, the point cloud represents the detected part, and the mesh is rendered by the analytic concept parameters. As shown in Tab. 8, all distances are notably small, with an average of 5.18 mm across train categories and 6.55 mm across test categories. These results underscore the accuracy of our parameter estimator in capturing both the structural and pose parameters of the objects.

Table 8: The first row shows the average accuracy of concept identification by the MLLM, valued in percentage. The second row shows the average distance from the point cloud of the detected part to the mesh rendered from the estimated analytic concept parameters, valued in millimeters.

| Evaluation | Training Categories | | | | | | | | | | | Testing Categories | | | | | |
|---|---|---|---|---|---|---|---|---|---|---|---|---|---|---|---|---|---|
| | Box | Dor | Fct | Fdr | Ket | Mcw | Stf | Swt | Tcn | Win | **AVG** | Bkt | Pot | Saf | Tab | Wsm | **AVG** |
| Accuracy | 97.5 | 95.0 | 91.7 | 82.6 | 96.4 | 92.9 | 96.7 | 94.8 | 94.8 | 90.0 | 94.2 | 97.6 | 88.0 | 95.9 | 97.8 | 88.9 | 95.6 |
| Distance | 9.24 | 3.66 | 1.92 | 1.85 | 2.65 | 5.67 | 5.42 | 1.84 | 9.80 | 8.62 | 5.18 | 9.73 | 3.41 | 9.75 | 8.65 | 12.4 | 6.55 |

## A.5 ANALYSIS OF GRASP POSE KNOWLEDGE

In Sec. 4.3 of the main paper, we have introduced grasp pose as an important type of analytic manipulation knowledge. Here, we quantitatively analyze the effectiveness of such knowledge.

We include a control group in which the grasp pose parameters are randomly *sampled* within the grasp pose parameter space, rather than being *estimated* by the grasp pose generator and discrim-

inator networks. The evaluation results are shown in Tab. 9. The comparison between baseline approach A3VLM and *sampled* demonstrates that our grasp pose knowledge in analytic concepts can precisely reveal the possible grasps for the robot gripper to interact with the target object. After employing our proposed neural networks to incorporate the 3D visual features (the point cloud $\mathcal{P}$ of the target part) for parameter estimation instead of relying on random sampling, the success rate of manipulations can be further improved, which demonstrates the effectiveness of our proposed grasp pose generator and discriminator networks.

Table 9: Evaluation results with grasp pose parameters randomly sampled in parameter space and estimated by our proposed neural networks. All values represent the average success rate as a percentage.

| Manipulation Knowledge | Training Categories | | | | | | | | | | | Testing Categories | | | | | |
|---|---|---|---|---|---|---|---|---|---|---|---|---|---|---|---|---|---|
| | Box | Dor | Fct | Fdr | Ket | Mcw | Stf | Swt | Tcn | Win | **AVG** | Bkt | Pot | Saf | Tab | Wsm | **AVG** |
| A3VLM | 12.2 | 56.2 | 24.1 | 40.8 | 20.1 | 49.7 | 53.3 | 25.4 | 18.2 | 15.8 | 37.4 | 25.2 | 11.0 | 40.0 | 39.1 | 22.3 | 32.1 |
| Sampled | 14.8 | 58.0 | 26.5 | 42.9 | 23.3 | 50.3 | 55.9 | 30.0 | 20.6 | 20.2 | 40.2 | 28.0 | 14.0 | 47.1 | 48.7 | 22.7 | 38.6 |
| Estimated | **15.9** | **58.4** | **28.0** | **46.7** | **24.3** | **52.2** | **58.6** | **32.7** | **22.9** | **22.3** | **42.5** | **28.9** | **15.5** | **50.6** | **51.6** | 22.7 | **40.8** |

To intuitively illustrate the difference between *estimated* and *sampled* grasp pose parameters, we provide examples of the manipulation knowledge "grasp around" applied to the analytic concept "*curved_handle*". As shown in Fig. 6-Left, the knowledge function is parameterized by two factors: 1) $\omega$, which defines the angle of the gripper turning along the handle, sampled within $(-\theta/2, \theta/2)$, where $\theta$ is the maximum allowable angle; 2) $\varphi$, which defines the angle of the gripper rotating around the handle, sampled within $(-\pi, \pi)$. In Fig. 6-Right, we provide ten examples of specific implementation of "grasp around" parameterized differently, and categorize them into four types. Grasp poses with blue background are physically plausible and are suitable to complete the task due to proper position and rotation. Grasp poses with green background are also physically plausible, but fail to hold the handle effectively. Grasp poses with yellow and red background are physically implausible, respectively colliding with the kettle itself and the tabletop (not shown in the figure). The grasp poses instantiated from *sampled* parameters could represent all possible candidates here, while only the parameters corresponding to the available grasp poses can be *estimated* by the grasp pose generator, which improves the effectiveness of grasping.

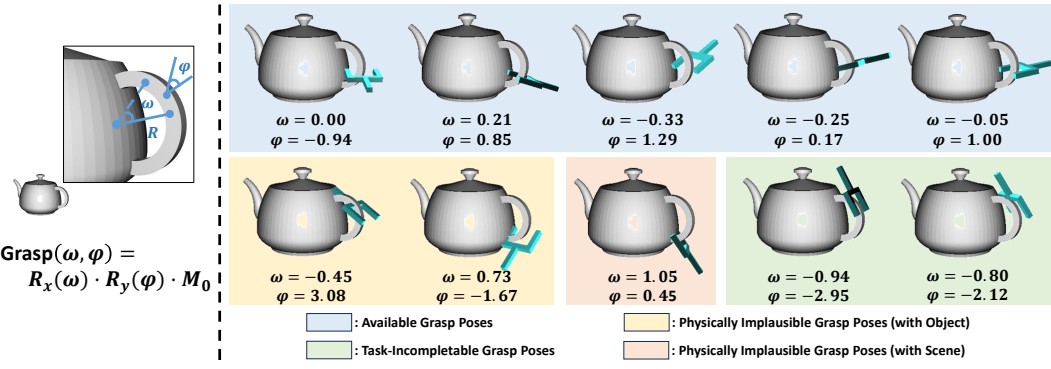

Figure 6: Examples of grasp poses for task *"pour the kettle"*. $\mathbf{M_0}$ denotes to the default pose of the gripper for grasping the handle, *i.e.* the grasp pose parameterized by $\omega = 0$ and $\varphi = 0$. $\mathbf{R_x}$ and $\mathbf{R_y}$ respectively represent rotations around the x-axis and the y-axis.

## A.6 ANALYSIS OF CONCEPT SYNOPSES IN PROMPTING

Here we provide an analysis on the effectiveness of synopses in prompting the MLLM to identify the concepts. We modify the synopses of analytic concepts to construct three controlled variants: 1) more descriptive synopses, 2) less descriptive synopses with sufficient semantic-level knowledge, and 3) less descriptive synopses lacking sufficient semantic-level knowledge. Specifically, we keep using GPT-4o (Hurst et al., 2024) as the MLLM and remain other modules the same as our current implementation. Tab. 10 shows the performance using synopses with variations of descriptiveness,

with "L_Handle" as an example. The results indicate that the concept identification accuracy and overall task performance remain peak performance and are barely affected as long as the synopses provide sufficient knowledge, while synopses with insufficient knowledge result in significant performance degradation.

Table 10: Evaluation results of effectiveness of synopses in prompting.

| | Concept Identification Accuracy | Overall Performance | Synopsis Example |
|---|---|---|---|
| *Current Implementation* | *94.2* | *42.5* | *A handle with a lever and an axis, perpendicular to each other, forming an L-shape. Can be turned around the axis, typically found on doors.* |
| More Descriptive | 95.0 | 42.7 | A handle consisting of a lever and a rotational axis arranged perpendicularly to form an L-shape. It allows turning around the axis and is commonly used for opening or closing doors. |
| Less descriptive (sufficient knowledge) | 93.5 | 42.2 | A handle with a lever and an axis perpendicular to each other, forming an L-shape. |
| Less descriptive (insufficient knowledge) | 52.8 | 22.5 | A type of handle forming an L-shape for turning. |

### A.7 DISTILLATION OF ANALYTIC CONCEPTS

We have so far created 153 analytic concepts, while 22 of them are used for manipulation tasks. Specifically, we iteratively distill redundant analytic concepts by analyzing their impact on manipulation success rates, retaining only those with effective contributions.

We initially select 15 object categories and invite volunteers to create analytic concepts to represent objects in these categories, along with associated manipulation knowledge. After cleaning and filtering, 153 analytic concepts are retained, which sufficiently cover all 15 categories. We then conduct experiments using these 153 concepts, as shown in **Set 1** of Tab. 11. However, many of these concepts are never identified by the MLLM during manipulation (*e.g. cuboidal_legs*, *cylindrical_body*). By removing unused concepts, we obtain **Set 2** with 59 concepts and observe close success rates, since the excluded concepts are not identified and thus causing no huge impact on performance. We further distill the 59 concepts to obtain **Set 3** through i) retaining only one concept for concepts representing highly similar parts (*e.g.* keeping *curved_handle* and removing *curved_tophandle*), and ii) removing concepts that can be composed by others (*e.g. symmetric_windows* can be constructed with multiple *regular_window*s). This progress has no significant impact on manipulation performance, as the knowledge of the removed concepts can still be captured by the remaining ones. In addition, we obtain **Set 4** with 22 concepts, *i.e.* the set we use for main experiments, by removing each of the 27 concepts individually, evaluating performance with the remaining concepts, and retaining only those whose removal causes significant performance drop. As a comparison, **Set 5** and **Set 6** respectively exclude the concepts representing lids/doors and handles/switches, showing significant drops in success rate, indicating that these concepts are crucial for completing manipulation tasks.

Table 11: Experiment results using different set of analytic concepts. Set 4 refers to the set of analytic concepts used in our main experiement.

| | Num of Concepts | Train Cat Results | Test Cat Results |
|---|---|---|---|
| Set 1 | 153 | 42.0 | 40.1 |
| Set 2 | 59 | 42.2 | 39.8 |
| Set 3 | 27 | 42.8 | 40.3 |
| *Set 4* | *22* | *42.5* | *40.8* |
| Set 5 | 17 | 35.4 | 29.8 |
| Set 6 | 7 | 10.3 | 6.4 |

## A.8 SCALABILITY OF ANALYTIC CONCEPTS

**Analytic Concepts' General Coverage on Real-world Objects.** As discussed in A.7, we use 22 analytic concepts to cover the actionable parts of 15 object types that are suitable for single-gripper manipulation in our experiments. We further analyze the number of concepts to cover the actionable parts of all 46 object categories in PartNet-Mobility (Xiang et al., 2020), an articulated object dataset containing real-world manufacturers' products. By randomly sampling 20, 30, and 40 categories ten times, the average numbers of required concepts are 27, 34, and 37 respectively. Covering the actionable parts of all 46 categories requires 39 concepts. This shows a diminishing marginal increase in required concepts as the object type grows. This stems from the fact that analytic concepts describe commonsense structural and manipulation knowledge of objects, which is reusable across different object categories. For example, the "L_Handle" concept can be used in object categories including door, storage furniture, faucet, window, etc.

**Scalability of Analytic Concepts to Novel Objects and Categories.** As mentioned above, the strong cross-category generalizability enables analytic concepts to easily scale to novel object types. Given a novel object type, we can first find reusable ones from the existing collection of analytic concepts. If the current collection is sufficient, scaling to new object types requires no additional effort. If the current collection is insufficient to fully represent the new object type, a small number of new concepts can be conveniently defined at program level through code inheritance and invocation from existing concepts and basic geometry templates.

**Scalability of Manually Constructing Analytic Concepts.** Manual construction of analytic concepts will not significantly hinder their scalability, because 1) only a few (sometimes even zero) new concepts are needed to cover a new object category, since analytic concepts are highly generalizable and reusable across different object categories, and 2) scaling up analytic concepts is practically convenient at program level through code inheritance and invocation from existing concepts and basic geometry templates, which enables developing a new concept in just about 2 hours on average as mentioned in Sec. 3.4. We will also explore the possibility of using algorithms to automatically develop new concepts as our future work, which is a promising way to drastically reduce human effort.

## A.9 ERROR DETECTION AND HANDLING

In robot manipulation experiments, we should detect and handle errors during the manipulation process to prevent unreasonable actions from damaging the robot and the target object. Taking advantage of analytic concepts and MLLMs in our propose approach, we are able to perform error detection and handling in an easy, flexible and effective way. Specifically, we **i)** predict the target part's state in advance according to the grounded structural knowledge and the determined manipulation knowledge, and **ii)** estimate the state of the target part during execution and compare it with the predicted one. A large discrepancy indicates an error which will trigger a stop signal. MLLM will be prompted to reconsider better results based on past failures.

## A.10 REAL WORLD DEMONSTRATIONS

We provide video demonstrations of real-world experiments in the supplementary material under the folder **experiment_videos**, which illustrate ten consecutive interactions on different types of articulated objects. We randomly change the pose of the objects in the workspace during the consecutive interactions to evaluate the robustness of our approach. The high success rate demonstrated in these videos highlights the effectiveness of our method in real-world scenarios.

### A.10.1 SIM-TO-REAL GAP

Although part of the training is conducted in a simulated environment, the gap between the simulated and real environments on our approach is relatively small for the following reasons: **1)** The Grounded-SAM (Ren et al., 2024) is trained on real data in real-world experiments to crop the target part's point cloud from the RGB image. **2)** The structure parameter estimation, 6-dof pose estimation modules are trained in a simulated environment and performed on the target part's point cloud. For a target part suitable for manipulation, typically a few centimeters in size, the point clouds captured in the real environment and the simulator do not differ significantly to produce drastically

different output for the modules. Thus the sim2real gap is rather small on the part-level. **3)** We adopt data augmentations to further narrow the sim2real gap, including adding noises and corruptions, and randomizing objects' poses and camera perspective.

### A.10.2 ROBUSTNESS IN REAL-WORLD EXPERIMENTS

In real-world experiments, we have identified several factors that may impact the effectiveness of our approach. We discuss about these factors in this section.

**Low-quality Point Cloud.** Although we strive for precise estimation of the target part's parameters and 6-dof pose, the primary focus of our work is to reasonably ground the knowledge inferred by MLLMs in the physical world, and finally enabling more accurate, stable, and controllable robot manipulation. Therefore, although the low-quality point cloud of the target part collected in real world in some cases may introduce slight inaccuracies in parameter and 6-dof pose estimation, our approach still ensures that the knowledge inferred by MLLMs is effectively grounded in physical world. As long as the estimated parameters and pose reasonably describe the target part, the robot can successfully complete manipulation tasks under the guidance by our approach.

**Similar Shapes.** As our study focuses on the spatial structure and function of target parts, we do not aim to fully represent all the details of the target part. While various real-world objects can share similar parts, they also share similar spatial structure and functions, which can be effectively captured by analytic concepts and manipulation knowledge. In this manner, the robot is guided to complete manipulation tasks on objects with similar target parts using similar knowledge. Thus similar shapes in target parts will not cause confusion.

### A.11 IMPLEMENTATION OF ANALYTIC CONCEPT

An analytic concept comprises three key components: concept identity, analytic structural knowledge, and analytic manipulation knowledge. In this section, we provide an example implementation of the analytic concept '**L_Handle**' in the form of Python class template.

### A.11.1 CONCEPT IDENTITY

Each analytic concept possesses a unique concept identity, *e.g.* '**L_Handle**'. Accompanying this identity is a precise and concise synopsis of the concept, designed to help MLLMs easily comprehend the commonsense knowledge represented by the concept.

```python
class L_Handle():
    """
    [synopsis] A handle with a lever and an
    axis, perpendicular to each other, forming
    an L shape. Can be turned around the axis,
    typically found on doors.
    """
```

### A.11.2 EXAMPLE OF BASIC GEOMETRY

Before introducing analytic structural knowledge, we first discuss about the basic geometry, which will be invoked in the implementation of analytic structural knowledge.

We present an example of a basic geometry '**Cuboid**'. This geometry class template applies shape deformations to the standard instance according to the given parameters, and finally returns the parameterized cuboid instance.

```python
class Cuboid():
    def __init__(self, size, position=[0, 0, 0], rotation=[0, 0, 0]):
        """
        Instantiate with parameters.
        """

        length = size[0]
```

```
1080          width = size[1]
1081          height = size[2]
1082
1083          # Register parameters
1084          self.length = length
1085          self.width = width
1086          self.height = height
1087          self.position = position
1088          self.rotation = rotation
1089
1090          # Manually defined standard template
1091          # instance, i.e. Cuboid with center at
1092          # [0, 0, 0], edge length 1, and edge
1093          # parallel to the coordinate axes.
1094          self.vertices = np.array([
1095              [-0.5,  0.5,  0.5],
1096              [ 0.5,  0.5,  0.5],
1097              [-0.5,  0.5, -0.5],
1098              [ 0.5,  0.5, -0.5],
1099              [-0.5, -0.5,  0.5],
1100              [ 0.5, -0.5,  0.5],
1101              [-0.5, -0.5, -0.5],
1102              [ 0.5, -0.5, -0.5]
1103          ])
1104          self.faces = np.array([
1105              [0, 1, 2], [1, 3, 2],
1106              [4, 6, 5], [5, 6, 7],
1107              [0, 4, 5], [0, 5, 1],
1108              [2, 7, 6], [2, 3, 7],
1109              [0, 6, 4], [0, 2, 6],
1110              [1, 5, 7], [1, 7, 3]
1111          ])
1112
1113          # Shape deformation
1114          vertices_resize = np.array([
1115              [length, width, height],
1116          ])
1117          self.vertices = self.vertices * vertices_resize
1118
1119          # Apply pose transformation to the
1120          # vertices
1121          self.vertices = apply_pose(
1122            self.vertices,
1123            position=position,
1124            rotation=rotation
1125          )
```

### A.11.3 ANALYTIC STRUCTURAL KNOWLEDGE

We now present an example implementation of analytic structural knowledge within analytic concepts. The code builds upon the previous section and continues within the class '**L_Handle**'.

Analytic structural knowledge employs mathematical procedure to capture the essential commonalities of the spatial structure within an analytic concept. Specifically, an analytic concept uses basic geometries (such as the '**Cuboid**' described earlier) to define its spatial structure. Parameters are introduced to represent variations across different physical instances. Given these parameters, the analytic concept generates its structure by parameterizing its basic geometries. The parameters for these basic geometries are calculated based on the provided inputs, adhering to the constraints embedded within the concept.

```
    def __init__(self, axis_length, axis_diameter, lever_length,
        lever_width, lever_height, axis_lever_offset, pose):
```

```
1134        # Register parameters
1135        self.axis_length = axis_length
1136        self.axis_diameter = axis_diameter
1137        self.lever_length = lever_length
1138        self.lever_width = lever_width
1139        self.lever_height = lever_height
1140        self.axis_lever_offset = axis_lever_offset
1141        self.pose = pose
1142
1143        # Instantiate component geometries
1144        axis_size = [axis_length, axis_diameter]
1145        axis_position = [
                -(axis_length + lever_width) / 2,
1146            -axis_lever_offset, 0
1147        ]
1148
1149        lever_size = [lever_length, lever_width, lever_height]
1150        lever_position = [0, 0, 0]
1151
1152        self.axis = Cylinder(axis_size, axis_position)
1153        self.lever = Cuboid(lever_size, lever_position)
1154
1155        # Apply the pose of the concept instance
1156        # to its component geometries
1157        self.axis.vertices = apply_pose(
              self.axis.vertices, pose=pose)
            self.lever.vertices = apply_pose(
              self.lever.vertices, pose=pose)
```

### A.11.4 ANALYTIC MANIPULATION KNOWLEDGE

We finally present an example implementation of analytic manipulation knowledge for analytic concepts. The code is integrated into the same class as described in the previous sections.

Analytic manipulation knowledge encompasses two key types: grasp poses and force directions, which guide appropriate interactions with a spatial structure. Each type may include multiple pieces of knowledge tailored to the concept. Similar to analytic structural knowledge, analytic manipulation knowledge is expressed through mathematical procedures representing shared features, along with variable parameters capturing variations. Additionally, a precise and concise synopsis is provided for each piece of knowledge, enabling MLLMs to understand these knowledge at semantic level.

For grasp poses, we define an initial pose and translate it to appropriate grasp positions as specified by the parameters in the concept's canonical space. Subsequently, we apply the concept's global transformation to the pose, allowing it to return the correct grasp pose in real-world space.

```
    def grasp_above(self, offset):
        """
        [synopsis] Gripper aims at the lever,
        approaches from above for grasping.
        """

        assert -self.axis_lever_offset <= \
          offset <= self.lever_length / 2

        # Default gripper pose
        initial_pose = Pose([[1, 0, 0, 0],
                             [0, 1, 0, 0],
                             [0, 0, 1, 0],
                             [0, 0, 0, 1]])

        # Move the gripper to target grasp pose
        # in the concept instance's canonical
        # space
```

```
1188          canonical_pose = initial_pose.translate(
1189            [0, offset, 0])
1190
1191          # Apply the concept instance's pose to
1192          # the canonical grasp pose
1193          real_pose = canonical_pose.apply(
1194            self.pose)
1195
1196          return real_pose
1197
1198      def grasp_front(self, offset):
1199          """
1200          [synopsis] Gripper aims at the lever,
1201          approaches from front for grasping.
1202          """
1203
1204          assert -self.lever_length / 2 <= \
1205            offset <= self.lever_length / 2
1206
1207          # Default gripper pose
1208          initial_pose = Pose([[1, 0, 0, 0],
1209                               [0, 1, 0, 0],
1210                               [0, 0, 1, 0],
1211                               [0, 0, 0, 1])
1212
1213          # Move the gripper to target grasp pose
1214          # in the concept instance's canonical
1215          # space
1216
1217          # 1. Rotate the gripper around y-axis
1218          # for 90 degrees
1219          tmp_pose = initial_pose.rotate(
1220            'y', 90)
1221
1222          # 2. Move the gripper along y-axis
1223          # according to offset
1224          canonical_pose = tmp_pose.translate(
1225            [0, offset, 0])
1226
1227          # Apply the concept instance's pose to
1228          # the canonical grasp pose
1229          real_pose = canonical_pose.apply(
1230            self.pose)
1231
1232          return real_pose
```

For force directions, we define several pieces of knowledge that specify the directions in which force should be applied to the target part during object manipulation. These directions are calculated with the attributes of the concept.

```
      def push_clockwise(self):
          """
          [synopsis] Gripper applies a force
          clockwise relative to the axis.
          """

          axis_direction = [-1, 0, 0]
          lever_direction = [0, 1, 0]

          # The force direction in the canonical
          # space is determined by the cross
          # product of the direction of
          # the lever and axis
```

```
        canonical_force_direction = cross_prod(
          axis_direction,
          lever_direction
        )

        # Apply the rotation of the concept
        # instance to the canonical force
        # direction for the real force
        # direction. Position is not required
        force_direction = apply_pose(
          canonical_force_direction,
          rotation=self.pose.rotation
        )

        return force_direction

    def pull_backward(self):
        """
        [synopsis] Gripper applies a force
        backwards relative to the handle.
        """

        axis_direction = [-1, 0, 0]

        # The force direction in the canonical
        # space is opposite to the direction
        # of the axis
        canonical_force_direction = \
          -axis_direction

        # Apply the rotation of the concept
        # instance to the canonical force
        # direction for the real force
        # direction. Position is not required
        force_direction = apply_pose(
          canonical_force_direction,
          rotation=self.pose.rotation
        )

        return force_direction
```

## A.12 GENERALIZATION CAPABILITY IN MANIPULATION TASKS

The generalization capability in manipulation tasks of our approach stems from two aspects: 1) highly expressive capability of analytic concepts to describe objects' structural knowledge, and 2) the atomic nature of manipulation knowledge. In addition to the discussions of these aspects, we also provide a discussion of how our approach handles manipulation tasks beyond common practice.

**Highly Expressive Capability of Analytic Concepts.** Analytic concepts can comprehensively describe the structural knowledge of various articulated objects encountered in daily life, as the design and manufacturing processes of man-made objects are based on combining a series of basic geometric primitives (*e.g.* CAD). Specifically, we introduce basic geometries (*e.g.* Cuboid, Cylinder) as the fundamental building blocks of analytic concepts to describe objects' structural knowledge. Each geometry encapsulates a category of shapes that share common spatial properties, while its associated parameters instantiate specific instances. Building on these basic geometries, analytic concepts describe an object's structural knowledge by combining these geometries, resulting in an exponential diversity in representing structural knowledge. As a reference, PartNet-Mobility (Xiang et al., 2020) is an articulated object dataset consisting of objects collected from 3D Warehouse, encompassing models of real-world manufacturers' products. In our experiments, our analytic concepts effectively describe the structural knowledge of various parts of objects from Xiang et al. (2020). The diver-

sity and complexity of the structural knowledge that analytic concepts can cover are comparable to real-world level, which demonstrates the highly expressive capability of our analytic concepts.

**Atomic Nature of Manipulation Knowledge.** The manipulation knowledge of robot articulated object manipulation tasks can be comprehensively defined due to the atomic nature of fundamental actions (*e.g.*, lift, turn, push, *etc.*) during manipulation, and these atomic actions are enumerable. An object manipulation process can be decomposed into a finite sequence of atomic actions (*e.g. opening a door* can consist of *grasping* the knob, *turning* it around its axis, and *pulling* the knob). By systematically combining these atomic actions, we can define a broad spectrum of manipulation knowledge to accommodate various object manipulation tasks. Leveraging the MLLM, our approach can determine the most reasonable manipulation knowledge for a specific manipulation task among the numerous manipulation knowledge that analytic concepts are able to cover. This enables robots to successfully complete a wide range of manipulation tasks.

**Beyond Common Practice.** The previous two properties enable our approach to cover a wide range of manipulation tasks on various articulated objects. However, there might be an exception where the robot is asked to perform a task that deviates from common practice (*e.g.* knock over a cup). Such a task poses a potential risk of danger and economic loss. As the robotics community has long focused on improving robot controllability to mitigate risks associated with unpredictable factors, the design of our method also follows this principle. When faced with tasks that go beyond common practice, our approach refrains from providing the robot with such manipulation knowledge, preventing it from engaging in potentially risky manipulation. In this manner, we can effectively ensure that the robot's behavior stays within reasonable bounds.

### A.13 LIMITATIONS AND FUTURE WORK

Our current work is a study of articulated object manipulation, through aligning semantic-level knowledge with physical-level knowledge. We focus on gripper-based manipulation of articulated objects and hence develop manipulation knowledge, aiming to validate the feasibility and effectiveness of using analytic concepts to improve the performance of articulated object manipulation tasks. In the future, we will extend our approach to support a broader range of end-effectors (*e.g.* dexterous hand), objects beyond articulated objects (*e.g.* deformable ones), complex scenerios (*e.g.* long-tasks involving multiple objects) to enable more comprehensive manipulation capabilities.

### A.14 EXPERIMENT COMPUTER RESOURCES

Our experiments are conducted on single NVIDIA A100 GPU and Intel(R) Xeon(R) Platinum 8276M CPU @ 2.20GHz.

### A.15 SOCIETAL IMPACTS

We propose analytic concepts as a bridge between semantic-level knowledge of articulated objects and the physical world, extending the applicability of MLLMs to real-world understanding and interaction, which may improve the quality of huamn life. However, this approach depends on MLLMs, which may incur additional economic and energy costs due to computation.

### A.16 STATEMENT ON USAGE OF LARGE LANGUAGE MODELS

This paper employs Large Language Models solely for polishing the writing, including grammar correction, phrasing refinement, and improvements in fluency and readability. No scientific ideas, experimental designs, results, or conclusions were generated by LLMs. All conceptual and technical contributions are entirely the work of the authors.

