# OpenReview forum: "Physically Ground Commonsense Knowledge for Articulated Object Manipulation with Analytic Concepts"
_ICLR.cc/2026/Conference — ICLR 2026 Conference Withdrawn Submission_

### Official Review · Reviewer_oPjT · 2025-10-27

**Soundness:** 2
**Presentation:** 2
**Contribution:** 3
**Rating:** 4
**Confidence:** 3

**Summary:**

The paper presents an approach that aims to bridge semantic knowledge and the physical world through analytic concepts, leveraging MLLMs for articulated object manipulation. While the topic is relevant and potentially valuable, the current manuscript suffers from serious issues in writing clarity, conceptual organization, and experimental rigor.

**Strengths:**

1. The general idea of connecting MLLM reasoning with robotic control through analytic concepts is interesting.
2. The motivation to achieve generalized manipulation is aligned with current trends in embodied AI.

**Weaknesses:**

1. The overall writing is weak and difficult to follow. Many sentences are unclear, and logical connections between sections are missing. The readability issues significantly hinder understanding of the main contributions.
2. Analytic Concepts Section is especially problematic. It is unclear why a “DISCUSSIONS” subsection appears here, this part should only provide definitions and conceptual clarifications.
3. Figures 1 and 2 are difficult to understand. They fail to clearly illustrate the framework or the flow of reasoning. Figure 3, which appears to represent the main method, is overly simplified and does not sufficiently explain the proposed approach.
4. The methodology section feels incomplete and fragmented, possibly due to space constraints. However, as it stands, the description is too brief to allow readers to follow the method.
5. The experimental setup is simple. From the limited results in Table 2 and Table 3, I cannot form a comprehensive judgment of the method’s performance. The experiments lack comparison and analysis with more meaningful baselines.

**Questions:**

1. How many data samples were used for training and evaluation in total?
2. Was your model trained from scratch or fine-tuned from an existing pretrained MLLM?
3. You mentioned using a single NVIDIA A100 GPU. What was your total training time?
4. Did you re-implement or reproduce the experimental results of other compared methods for fair comparison?

---

### Official Review · Reviewer_zCrC · 2025-10-29

**Soundness:** 1
**Presentation:** 1
**Contribution:** 2
**Rating:** 2
**Confidence:** 3

**Summary:**

This paper proposes using analytic concepts to represent commonsense knowledge via MLLMs, and then grounding them in physical form so that robots can complete manipulation tasks more easily.

**Strengths:**

The idea of introducing analytic concepts and connecting them to physical actions seems interesting.

**Weaknesses:**

1. The writing needs significant improvement. The current version is very hard to read and follow. Too much information is pushed to the appendix, while the main paper lacks concrete examples. The authors try to present too many things, but the content is not well organized. For acceptance at a top AI venue, the paper should be made clearer and more structured.

2. Figures: there is no main figure that clearly shows the overall workflow. Fig. 1(b) at the bottom explains it briefly, but this is insufficient (and it is not even clearly indicated that it is from the authors). In addition, the fonts are too small in most figures (it is hard to read the text in a printed version)

3. It is difficult to understand what differentiates the proposed method from the baselines. Even though related works are mentioned, the contribution is not clear. Much of Section 4 (Methodology) simply presents components of the method without motivation or clear comparison to prior work

4. The explanation of analytic concepts is unclear. Similar structure-based representations have been used for many robotics tasks [1]. What makes the proposed concepts novel? Why are they called “analytic”? The authors mention that detailed examples are provided in the appendix, but a simple and concrete example should be included in the main paper for clarity.

[1] Samuel Li et al.,"ShapeGrasp: Zero-Shot Task-Oriented Grasping with Large Language Models through Geometric Decomposition," IROS 2024

**Questions:**

See Weaknesses

---

### Official Review · Reviewer_RY1Z · 2025-10-30

**Soundness:** 3
**Presentation:** 2
**Contribution:** 2
**Rating:** 2
**Confidence:** 3

**Summary:**

This paper presents a framework for robotic manipulation that combines multi-modal large language models (MLLMs) with analytically defined manipulation concepts—symbolic templates that encode structural and interaction priors for articulated objects. These concepts are manually crafted to reflect physically grounded properties of common object parts (e.g., handles, lids) and their associated manipulation strategies. The MLLM selects the appropriate concept and strategy, while learned perceptual modules extract relevant physical parameters from point-cloud observations. Experiments in both simulation and real-world scenarios demonstrate improved performance over prior MLLM-based approaches (e.g., A3VLM, ManipLLM).

**Strengths:**

- **Strong balance between LLM reasoning and grounded control.**
  The paper effectively bridges high-level LLM-based affordance reasoning with physically grounded manipulation strategies, offering a compelling middle ground between symbolic priors and data-driven policy learning.

- **Solid generalization performance.**
  The reported 27.1% gain on unseen object categories, along with consistent performance across both suction-based and parallel-jaw gripper setups, demonstrates strong generalization and robustness.

- **Transparent and interpretable pipeline.**
  The system's modular structure allows clear diagnosis of failure modes (e.g., pose estimation vs. analytic concept selection), making it easier to analyze and improve individual components.

- **Practical and accessible analytic concept design.**
  The authors convincingly argue that analytic concepts are simple to author—even for non-experts—and that a relatively small library (153 total, with only a subset needed per environment) can cover a wide range of real-world tasks.

**Weaknesses:**

- **Dependence on analytic concept library.**
  While the analytic concepts are elegant, the approach fundamentally relies on the completeness and correctness of the concept library. The paper argues that these concepts are easy for humans to author, but it remains unclear how scalable and generalizable this process is in practice. For instance, how does the system handle objects or interaction modes that do not align cleanly with any existing concept? The method implicitly assumes that the library is sufficiently comprehensive, which may not always hold in real‐world deployments.

- **Assumptions about perception reliability.**
  The pipeline assumes that the MLLM and Grounded-SAM can consistently identify the correct part and category from a single RGB image. In realistic scenarios, occlusions, ambiguous language cues, or objects with multiple plausible interaction points may challenge this assumption. Additional analysis or experiments examining such failure cases would strengthen the paper.

- **Limited evaluation scope for task complexity.**
  Experiments are primarily conducted on single-step tasks with a single type of gripper. It remains unclear how analytic concepts extend to more complex settings, such as multi-stage manipulation (e.g., opening a door before pulling a drawer), deformable objects, or tool‐use scenarios. Although the proposed method appears conceptually compatible with long-horizon tasks, demonstrating results beyond single-step manipulation would provide a more complete understanding of the method’s practical capabilities.

**Questions:**

- **Coverage and fallback mechanisms.**
  Although the paper argues that analytic concepts are straightforward to author, the method assumes that the concept library is sufficiently comprehensive. How does the system behave when encountering objects that only partially align with existing concepts or fall entirely outside the library? Is there a fallback mechanism—such as compositional reasoning, interpolation between concepts, or automatic discovery of new concepts—or does performance degrade sharply in these out-of-distribution scenarios?

- **Extension to long-horizon and sequential manipulation.**
  The experiments mainly evaluate single-step tasks with a single gripper modality. Can analytic concepts be composed to support multi-stage, long-horizon manipulation with sequential dependencies (e.g., open lid → insert spoon → stir)? Do the authors envision the framework performing hierarchical reasoning and re-grounding at each step, or is it currently limited to feed-forward, single-stage execution?

- **Generalization beyond rigid articulated objects.**
  The framework is currently tailored to rigid, articulated objects. Do the authors view analytic concepts as a general representation for physical commonsense that could extend to deformable objects, tool-use, or non-rigid manipulation? If so, what modifications would be required to accommodate more complex material properties and dynamic interactions?

- **Failure mode analysis and comparative insights.**
  The paper would benefit from a more detailed failure-mode analysis. What are the most common sources of error observed in practice? Additionally, highlighting scenarios where the proposed analytic-concept approach succeeds compared to prior methods would help clarify its practical strengths and limitations for real-world deployment.

---

### Note · Authors · 2025-11-13

I have read and agree with the venue's withdrawal policy on behalf of myself and my co-authors.